# Optimisation of *β*-Glucosidase Production in a Crude *Aspergillus japonicus* VIT-SB1 Cellulase Cocktail Using One Variable at a Time and Statistical Methods and its Application in Cellulose Hydrolysis

**DOI:** 10.3390/ijms24129928

**Published:** 2023-06-09

**Authors:** Nivisti Singh, Bishop Bruce Sithole, Roshini Govinden

**Affiliations:** 1Discipline of Microbiology, School of Life Sciences, Westville Campus, University of KwaZulu-Natal, Durban 4001, South Africa; nivistisingh@gmail.com; 2Discipline of Engineering, Howard Campus, University of KwaZulu-Natal, Durban 4001, South Africa; sitholeb1@ukzn.ac.za

**Keywords:** cellulases, *β*-glucosidase, endoglucanase, exoglucanase, Plackett Burman design, Box Behnken design, cellulose hydrolysis

## Abstract

Pulp and paper mill sludge (PPMS) is currently disposed of into landfills which are reaching their maximum capacity. Valorisation of PPMS by enzymatic hydrolysis using cellulases is an alternative strategy. Existing commercial cellulases are expensive and contain low titres of *β*-glucosidases. In this study, *β*-glucosidase production was optimised by *Aspergillus japonicus* VIT-SB1 to obtain higher *β*-glucosidase titres using the One Variable at a Time (OVAT), Plackett Burman (PBD), and Box Behnken design (BBD)of experiments and the efficiency of the optimised cellulase cocktail to hydrolyse cellulose was tested. *β*-Glucosidase production was enhanced from 0.4 to 10.13 U/mL, representing a 25.3-fold increase in production levels after optimisation. The optimal BBD production conditions were 6 days of fermentation at 20 °C, 125 rpm, 1.75% soy peptone, and 1.25% wheat bran in (pH 6.0) buffer. The optimal pH for *β*-glucosidase activity in the crude cellulase cocktail was (pH 5.0) at 50 °C. Optimal cellulose hydrolysis using the crude cellulase cocktail occurred at longer incubation times, and higher substrate loads and enzyme doses. Cellulose hydrolysis with the *A. japonicus* VIT-SB1 cellulase cocktail and commercial cellulase cocktails resulted in glucose yields of 15.12 and 12.33 µmol/mL glucose, respectively. Supplementation of the commercial cellulase cocktail with 0.25 U/mg of β-glucosidase resulted in a 19.8% increase in glucose yield.

## 1. Introduction

The pulp and paper-making process produces large quantities of solid waste referred to as pulp and paper mill sludge (PPMS). Four to five hundred thousand wet tonnes of PPMS are produced in South Africa alone, making the industry one of the highest polluters [1]. For many years, PPMS has been disposed of into landfills or incinerated. Escalating global environmental awareness, financial concerns, and landfills reaching their maximum capacity makes investigation of alternative methods imperative [2]. One such method includes the valorisation of PPMS into value-added products; firstly, through the recovery of fermentable sugars such as glucose, this can serve as feedstock for the production of valuable chemicals and pharmaceuticals. For example, the biofuel industry is dependent on lignocellulosic residues as an abundant and renewable energy source. Chemical and enzymatic methods are used to hydrolyse the cellulose polymer into fermentable sugars which can then be converted to bioethanol [3]. Chemical methods were preferred; however, the harmful substances released exacerbate environmental pollution, whilst enzymatic methods are considered a “green” method with a reduced impact on the environment [4].

The majority of PPMS is composed of lignocellulosic residues originating from the raw material, wood with (25%) lignin and holocellulose, consisting of (25%) hemicellulose and (50%) cellulose [5]. Cellulose is a linear homopolymer composed of _D_-glucose monomers, which are linked by *β*-1-4 glycosidic bonds [6] and is embedded in hemicellulose and lignin; both of which limit the rate of utilisation of untreated biomass [7]. Cellulases are a group of enzymes including endoglucanases, exoglucanases, and *β*-glucosidases, that are responsible for the synergistic hydrolysis of cellulose into oligosaccharides, cellobiose, and glucose [8]. Each enzyme displays a distinct function; endoglucanase and exoglucanase act within and on the cellulose chain ends, respectively releasing cellobiose, cello-oligosaccharides, and some glucose molecules [4]. In the final step, *β*-glucosidases break down cellobiose and cello-oligosaccharides into glucos, resulting in the complete degradation of cellulose [8].

During lignocellulosic biomass conversion in the biofuel industry, cellobiose acts as an inhibitor of exoglucanase and endoglucanase. Cellobiose is hydrolysed by *β*-glucosidase. However, the activity of *β*-glucosidases is the rate-limiting step as they in turn are subject to feedback inhibition at high glucose concentrations; therefore, glucose accumulation results in feedback inhibition and reduced glucose yields [4]. Thus, *β*-glucosidase is incorporated to mitigate glucose inhibition resulting from cellobiose hydrolysis. Feedback inhibition occurs due to transglycosylation events due to the activity of *β*-glucosidases being a reversible process, in which the enzyme utilises the active site to form a glycosidic bond instead of hydrolysing it [9]. The main obstacles to an efficient industrial cellulolytic process centre around the cellulase cocktail, including challenges such as low *β*-glucosidase titres, sensitivity to and inhibition by the product glucose, and thermal inactivation, resulting in low glucose yields, as well as the high cost of enzymes [10].

Species of *Aspergillus* and *Trichoderma* are currently in the spotlight for industrial applications due to their capacity to produce multiple extracellular enzymes; however, there is a need for a strain with highly functional endoglucanases, exoglucanases, and *β*-glucosidases in optimal ratios [11]. Current *T. reesei* cellulase cocktails utilised in industry lack sufficient *β*-glucosidases for efficient cellulose hydrolysis and require supplementation of *β*-glucosidases [12]. Rani et al. [13] reported hydrolysis of cellulose by a commercial *T. reesei* cellulase cocktail with a ratio of <1 *β*-glucosidase: 18 endoglucanase 72 exoglucanase. However, the reaction had to be supplemented with *β*-glucosidases from *A. niger*. In another, more recent study, Darwesh et al. [1] isolated *A. niger* strain MK543209 with a cellulase ratio of 1 *β*-glucosidase: 1.5 endoglucanase: 1.3 exoglucanase, which yielded 0.9 g/L ethanol from PPMS.

A cellulase cocktail with optimal ratios of the three cellulase enzymes, specifically higher *β*-glucosidase titres, is key for efficient cellulose hydrolysis. Optimisation of fermentation conditions is also crucial for profitable enzyme production for commercialisation. Optimisation methods include One variable at a time (OVAT), Placket Burman design (PBD), and Box Behnken design (BBD). During OVAT optimisation, one variable at a time is changed whilst all the other factors are kept constant. However, this method ignores the interactions between different independent parameters and is laborious, time consuming, and costly. Statistical methods such as PBD and BBD take into consideration the interactions between the variables and require a smaller number of experiments. Nisar et al. [14] obtained a 68% increase in *β*-glucosidase production by a *Thermomyces dupontii* mutant to 36 U/mL using PBD and BBD. In another study using BBD, 31.1 U/mg of *β*-glucosidase activity was obtained from *A. terreus* [15].

Several studies have reported the production of multiple cellulase enzymes from species of *Aspergillus* [1,7,15]. Mishra and Suneetha [16] first isolated *A*. *japonicus,* a type of mould that survives in tropical and subtropical outdoor environments, as well as submerged organic debris in 2014. The morphology of this isolate ranges from distributed mycelia to compacted pellets with purplish black hues [16].

The current study was set out to optimise *β*-glucosidase production in a crude *A. japonicus* VIT-SB1 cellulase cocktail produced under submerged fermentation using OVAT, PBD, and BBD and to assess the efficiency of the optimised cellulase cocktail on cellulose hydrolysis. Bioprospecting and optimisation studies for *β*-glucosidases are relevant, as there is a dearth of research regarding improving the titres in commercial cellulase cocktails.

## 2. Results and Discussion

### 2.1. One Variable at a Time (OVAT)

Commercial cellulase cocktails currently used for the hydrolysis of cellulose comprise low titres of *β*-glucosidases and result in low glucose yields [17]. The gap in research lies in searching for a cocktail of cellulases with higher *β*-glucosidase titres from native microbial producers [3]. A native strain producing high-activity cellulases with higher *β*-glucosidase titres relative to endoglucanase and exoglucanase may produce higher glucose yields. This will be valuable to industry, as the commercial cocktails formulated from individual pure cellulases are expensive. Species of *Aspergillus* and *Trichoderma* are in the spotlight, as they display great potential for application in biotechnological processes [18]. The above-mentioned factors were the impetus to this study on the stepwise optimisation of *β*-glucosidase production by *A. japonicus* VIT-SB1 in a process using one variable at a time experiments, as well as the statistically designed experiments, Plackett Burman (PBD) and Box Behnken (BBD). During OVAT experiments, endoglucanase and exoglucanase production were also monitored to select parameters that permitted their production concomitant with enhanced *β*-glucosidase production. Optimal *β*-glucosidase production was obtained on day six in sodium acetate buffer (pH 5.0) at 30 °C and an agitation speed of 125 rpm (Figure 1a–d). Kanakaraju et al. [19] reported optimal *β*-glucosidase production from *Aspergillus* species DHE7 on days five and six in accordance with the results in the current study. El-Nagger et al. [15] obtained optimal *β*-glucosidase production between (pH 3.5 and 5.0) at 30 °C from the *A. terreus* strain EMOO 6-4 grown in 1.25% wheat bran and 1.75% soy peptone as carbon and nitrogen sources, respectively (Figure 1e–h). Wheat bran has been reported to be the carbon source of choice as it is not only a good inducer of cellulases and hemicellulases, but is also cheap. The complex structure of wheat bran makes it difficult for filamentous fungi to penetrate the structure, and as a result it produces more enzymes to aid penetration [20]. Vaithanomsat et al. [21] and Abdullah et al. [20] reported optimal *β*-glucosidase production by *A. niger* and *A. oryzae*, respectively, on wheat bran.

Following OVAT optimisation experiments, *β*-glucosidase levels in the crude extract increased from 0.4 U/mL to 2.25 U/mL, representing a 5.6-fold increase in *β*-glucosidase production. Endoglucanases (12.5 U/mL) and exoglucanases (7.8 U/mL) were also detected at conditions selected for optimal *β*-glucosidase production. Silva-Mendoza et al. [22] also observed optimal endoglucanase and exoglucanase production in minimal media supplemented with wheat bran over a wide pH (4.0–6.0) and temperature (30–70 °C) range over longer periods (5 days).

### 2.2. Optimisation of Significant Variables for β-Glucosidase Production

The effect of six parameters on *β*-glucosidase production was studied using the Plackett Burman Design (PBD). The high (+) and (−) low levels were selected based on the optimal (0) levels obtained from the OVAT experiments to confirm that the high and low were equidistant from the optimal level and adequately far apart from one another, and to ensure that both endoglucanase and exoglucanase production was permitted at all three levels selected for PBD and BBD (Section 3.5.1. The rows in Section 3.5.1 represent the 12 experimental runs displaying *β*-glucosidase activities ranging from 0.01 to 7.03 U/mL across the 12 experimental runs. A further 3.11-fold increase in *β*-glucosidase activity was obtained following PBD compared to the activity after the OVAT experiments. The pH (X_2_) and incubation temperature (X_3_) significantly improved *β*-glucosidase production *p* < 0.05) (Table 1). The model, agitation (X_4_), and soy peptone (X_6_) displayed *p*-values < 0.1, indicating significance at a 90% confidence level. A 90% confidence interval has been used before; however, in the medical and pharmaceutical industry, this is not acceptable [10].

Higher R^2^ values indicate a better fit, as the model is parallel to observations; however, higher R^2^ values do not necessarily mean that the model suitably expresses the relationship between the predictors and responses, as any effect added to the model including noise will increase the R^2^ value. Therefore, the R^2^ value (0.83) obtained in this study is acceptable [23]. Figure 2a indicates that the actual values were close to the predicted values. The Pareto chart of standardisation (Figure 2b) confirmed that pH (X_2_), incubation temperature (X_3_), and agitation (X_4_) significantly influenced *β*-glucosidase production (*p* < 0.1), as they crossed the *p*-line. The other independent variables (*p* > 0.1) were considered insignificant. The regression equation was used to express the corresponding response to *β*-glucosidase activity using the unstandardised Beta values
(1)$$Y=X_{1}+X_{2}+X_{3}+X_{4}+X_{5}+X_{6}\;Y=15.093-0.68X_{1}-1.28X_{2}+1.28X_{3}-0.4X_{4}+3.99X_{5}+0.18X_{6}$$
where Y is defined as the peak area, X_1_ refers to the incubation time, X_2_ is the incubation temperature, X_3_ is the pH, X_4_ is the agitation, X_5_ is the soy peptone, and X_6_ is the wheat bran.

### 2.3. Box Behnken Design for β-Glucosidase Production

A 16 run Box Behnken Design (BBD) was used to determine the optimal conditions for *β*-glucosidase production by *A. japonicus* VIT SB1. The three significant variables pH, incubation temperature, and agitation (Section 3.5.2) were used to generate the BBD matrix (Section 3.5.2). The R^2^ or coefficient of determination of a valid model should be close to 1. The R^2^ value of 0.98 obtained validated the model, indicating that 98% of the variability of the response can be explained by the model [24]. The adjusted R^2^ (0.97, close to 1) confirms that the actual values were close to the predicted values, which can be confirmed in Figure 3a that displays the correlation between the actual and predicted values [25]. The Pareto chart of standardisation (Figure 3b) confirmed that incubation temperature (X_3_), significantly influenced *β*-glucosidase production (*p* < 0.05), as they crossed the *p*-line.

The maximum *β*-glucosidase activity (10.13 U/mL) in the BBD for *A. japonicus* VIT-SB1 was observed in run 2 with the following conditions: sodium acetate buffer (pH 6.0), lower incubation temperature (20 °C), lower agitation (125 rpm), and supplementation with optimal concentrations of soy peptone (1.75%) and wheat bran (1.25%). Lower activity was observed in run 11 (8.81 U/mL) in minimal media supplemented with optimal concentrations of carbon/nitrogen in sodium phosphate buffer (pH 5.0), at 20 °C and 150 rpm. Even lower activities were observed in run 6 (3.36 U/mL) and 9 (3.46 U/mL) sodium acetate buffer (pH 6.0 and 5.0), at incubation temperatures of 30 and 20 °C, and an agitation of 100 rpm. This may be an indication of the presence of *β*-glucosidase isoforms in the *A. japonicus* VIT-SB1 crude extract. Isoform production is induced by different incubation times, temperatures, and carbon sources [18]. The presence of *β*-glucosidase isoforms in fungal crude extracts was reported in many enzyme optimisation studies [26,27,28,29].

The model is significant (*p* < 0.05) (Table 2). The lack of fit *p*-value was greater than 0.05, thus indicating that the model fitted well. Lack of fit *p*-values greater than 0.05 were reported by several researchers [14,15,30]. The model, the linear term for (X_3_) incubation temperature, and square terms for X_4_ (agitation) were significant with *p*-values of 0.04, 0.0003, and 0.04, respectively (Table 2). The interactions between all 3 variables were significant with *p*-values < 0.05 (Table 2). The second-order regression equation provides the *β*-glucosidase activity produced by *A. japonicus* VIT-SB1 as a function of pH (X_2_), incubation temperature (X_3_), and agitation (X_4_), which are presented in the following Equation (2):Y = −2.58 + 1.293X_2_ + 7.64X_3_ − 1.59X_4_ + 1.66X_2_^2^ + 58.37X_3_^2^ − 2.53X_3_^2^ + 9.86X_23_ − 2.05X_24_ − 12.15X_34_(2)

### 2.4. Interaction of Variables for β-Glucosidase Production

The relationship between the responses and the parameters for *β*-glucosidase activity generated by the quadratic model and the optimum level of each variable for *β*-glucosidase activity is studied by the three-dimensional (3D) response surface plots (Figure 4a–c). The z-axis in the 3D response surface plots refers to *β*-glucosidase activity versus any two variables, whilst the other variables are at their optimal levels. Both plots display a range of values for each variable at which optimal enzyme activity may be obtained. Figure 4a1,a2 illustrates the 3D response surface and contour plots, respectively, for the interaction effects of the combined variables pH and agitation. *β*-Glucosidase activity was directly proportional to pH and inversely proportional to agitation (Figure 4a1), with the contour plot indicating high enzyme activity obtained at the lowest agitation (100 rpm) and highest agitation (140 rpm) in acidic conditions (pH 5.0–6.0) (Figure 4a2). On the other hand, the interaction effects of pH and incubation temperature (Figure 4b1,b2) reveal an inversely proportional relationship between temperature and enzyme activity and a directly proportional relationship with pH, with the highest activity at (pH 6.0) and the lowest incubation temperature (20 °C). Two studies by Mahapatra et al. [30] and Song et al. [31] obtained high *β*-glucosidase production at lower temperatures between 20 and 30 °C, at (pH 5.0 and 6.0), respectively. Figure 4c1,c2 displays the interaction between agitation and incubation temperature. The interaction between agitation and temperature demonstrates that high *β*-glucosidase activity was obtained at high agitation and low incubation temperature (Figure 4c1). The contour plots imply that the highest *β*-glucosidase activity was obtained at high agitation (125 rpm) and the lowest incubation temperature (20 °C) (Figure 4c2).

The 3D plots and the contour plots correlate with the results of the analysis of variable (ANOVA) table, which indicate that the interactions between the 3 variables are significant with *p*-values < 0.05 (Table 2). These results are corroborated by several other studies that reported higher *β*-glucosidase production at acidic pH and lower incubation temperatures [14,31,32,33]. This is not surprising, as temperatures between 25 and 30 °C and (pH 6.0) are reported to be excellent conditions for the growth of mesophilic fungi [33,34]. The interaction between pH and incubation temperature (9 U/mL) displayed the highest effect on *β*-glucosidase production compared to the other two interactions between pH and agitation (3.5 U/mL) and agitation and incubation temperature (8 U/mL) (Figure 4a–c).

This study demonstrated a notable increase in *β*-glucosidase production using the statistically designed experiments when compared to OVAT. A 48.5-fold increase in *β*-glucosidase production was observed after sequential optimisation of the *A. japonicus* VIT-SB1crude extract. *β*-Glucosidase activity improved from an original titre of 0.4 U/mL obtained in a previous study by Singh et al. [35], to 2.25 U/mL in OVAT, to 7.03 U/mL in PBD, and finally, to 10.13 U/mL using the BBD. Multiple studies utilise cellulose, avicel, cellobiose, and glucose as a carbon source for induction of cellulase production; however, cellobiose and glucose result in low *β*-glucosidase production levels whilst cellulose and avicel enhance it [15,34,35,36].

Lee et al. [34] utilised 3% cellulose as a carbon source which was supplemented with 0.1% thiamine to obtain a maximal *β*-glucosidase activity of 42.3 U/mL from *Schizophyllum commune* KUC939. They achieved higher levels of *β*-glucosidase production compared to the current study. However, although a small amount of thiamine was supplemented into the fermentation and may be worth the expense to obtain higher enzyme yields, a relatively high concentration of cellulose was also utilised. Cellulose is extremely expensive and will result in escalated financial costs for larger scale production, rendering this approach to be impractical and the use of commercial enzymes to remain the best option for industries [17]. Although the maximal *β*-glucosidase activity of 10.13 U/mL obtained in this study is approximately 4 times lower than that obtained by Lee et al. [34], this and supplementation of commercial cellulase cocktails with *β*-glucosidases may still be a more feasible option for industries, as wheat bran is a cheaper carbon source option (less than one-third of the cost of cellulose) [15].

### 2.5. Scaled-Up Fermentation in Optimised Conditions

The *β*-glucosidase enzyme production was scaled-up in shake flask fermentations for further studies. Each 2 L flask contained 400 mL of the minimal media (pH 6.0) supplemented with optimal nitrogen (1.75% soy peptone) and carbon (1.25% wheat bran) concentrations established during OVAT, and incubated at optimal conditions (20 °C for 6 days shaking at 125 rpm). The enzyme activity was determined as described previously and compared to those obtained at a smaller scale. Enzyme titres of 10.21 U/mL were obtained, similar to those obtained in the smaller scale fermentations in run 2 (Section 3.5.2. The initial crude *A. japonicus* VIT-SB1 cellulase cocktail displayed a ratio of 1:88:155 of *β*-glucosidase: endoglucanase: exoglucanase. The final cellulase activities obtained in the optimised scaled-up crude extract from *A. japonicus* VIT-SB1 were *β*-glucosidase (10.21 U/mL), endoglucanase (1.4 U/mL), and exoglucanase (2.4 U/mL), translating to a ratio of *β*-glucosidase: endoglucanase: exoglucanase (1:0.1:0.2) (Figure 5). The dramatic shift in ratios of each cellulase from the initial crude extract obtained in the study by Singh et al. [35] can be attributed to the optimal high and low levels selected based on conditions that favoured optimal *β*-glucosidase production. Therefore, the conditions selected resulted in reduced endoglucanase and exoglucanase production and increased *β*-glucosidase production.

### 2.6. Characterisation of the Crude Extract Produced by A. japonicus VIT-SB1

#### 2.6.1. Polyacrylamide Gels

Analysis by SDS-PAGE revealed multiple bands ranging from approximately 250 to less than 15 kDa in molecular mass (Figure 6). Native-PAGE confirmed the presence of all three cellulases in the *A. japonicus* VIT-SB1 crude extract. Figure 6 lane 2 reveals more than one area of black precipitation indicative of *β*-glucosidase activity in the *A. japonicus* VIT-SB1 crude extract. A large area of blackening indicates *β*-glucosidases ranging from 250 to 130 kDa and a single band of blackening at 100 kDa. In this assay, *β*-glucosidase hydrolyses esculin to glucose and esculetin; the esculetin reacts with ferric ions, producing the black precipitate, and confirming the presence of *β*-glucosidase isoforms in the crude extract. Several other studies reported *β*-glucosidase isoforms in the crude extracts [10,17,28,36]. Zymogram analyses with carboxymethylcellulose and avicel confirmed the presence of endoglucanase and exoglucanase isoforms, respectively. Figure 6 lane 3 demonstrates endoglucanases ranging in molecular mass from 130 to 77 kDa presented by the clearing of the entire lane and a single band of clearing of approximately 40 kDa. Exoglucanase also displays clearance of the entire lane in Figure 6 lane 4, revealing exoglucanases ranging in molecular mass (130–77 kDa). The zones of clearance after Congo red staining demonstrate the hydrolytic activity of endoglucanases and exoglucanases on cellulose to cellobiose and smaller sugar units. A study by den Haan et al. [37] reported the presence of endoglucanase and exoglucanases of similar molecular mass.

#### 2.6.2. Optimum pH, Temperature, and Stability

Activity peaks are evident at (pH 3.0 and 5.0), indicating the presence of *β*-glucosidase isoforms; however, maximum activity of the *A. japonicus* VIT-SB1 *β*-glucosidase was at pH 5.0 (Figure 7a). At (pH 3.0), the enzyme lost more than 20% of its activity within the first 30 min, then remained stable between 60 and 180 min. At (pH 5.0), 100% *β*-glucosidase activity was retained within the first 30 min; at 60 min, 10% activity was lost, after which the enzyme remained stable for up to 120 min. A further 10% of activity was lost between 120 and 180 min (Figure 7b). The enzyme displayed greater stability at (pH 5.0) compared to (pH 3.0), and therefore, the optimal temperature of the enzyme was determined at (pH 5.0). In addition, the optimal pH range for cellulose hydrolysis is (pH 5.0–9.0) [5]. At (pH 5.0), optimal temperatures of 50 and 85 °C were obtained (Figure 8a). At 85 °C, the *β*-glucosidase enzyme was unstable and lost 60% activity within the first 30 min (Figure 8b). A further 20% of activity was lost at 60 min, and at 180 min only 10% of activity was retained (Figure 8b). The enzyme was stable at 50 °C, signifying that this is a thermophilic *β*-glucosidase enzyme (Figure 8b). Sulyman et al. [38] also reported an optimal temperature of 50 °C for *β*-glucosidase at (pH 5.0); however, they did not determine the stability of the enzyme. Endoglucanase and exoglucanase activity were quantified in the crude cellulase cocktail at optimal β-glucosidase conditions (pH 5.0) and (50 °C) indicating that both the enzymes are thermophilic [2].

### 2.7. Cellulose Hydrolysis by BBD

This study focused on conditions favouring *β*-glucosidase activity, as studies report that this enzyme is the rate-limiting enzyme in cellulose hydrolysis [13,17,28]. Therefore, an optimal (pH 5.0) and temperature (50 °C) favouring the highest *β*-glucosidase activity in the crude *A. japonicus* cellulase extract were selected as the ideal conditions for the hydrolysis of cellulose (Figure 7 and Figure 8). The model was significant (*p* < 0.05) (Table 3). The lack of fit *p*-value was 0.39, indicating that the model fitted well. An R^2^ value of 0.95 and an F-value of 27.18 validated the model. The adjusted R^2^ value of 0.92 (R^2^ close to 1) confirms that 92% of the values obtained can be explained by the model’s response and indicates that the predicted and actual values are in close proximity. The linear terms for (X_1_) incubation time, (X_2_) incubation temperature, (X_3_) substrate load, (X_4_) enzyme dose, and the interactions between (X_1_) incubation time and (X_4_) enzyme dose, and square terms of (X_1_) incubation time, (X_2_) incubation temperature, and X_4_ enzyme dose were significant, with *p*-values < 0.05 (Table 3). The Pareto chart analysis confirms that the results obtained in ANOVA are significant, with the three most significant terms being the linear term for (X_2_) incubation temperature, the interaction between (X_1_) incubation time and (X_4_) enzyme dose, as well as the square term for (X_4_) enzyme dose (Figure 9).

The highest hydrolysis was obtained in run 16 with a glucose yield of 15.24 µmol/mL, followed by run 13 with 11.66 µmol/mL, and run 7 with 10.16 µmol/mL glucose Section 3.11). Run 16 glucose yields were obtained after the longest incubation time (120 min) and highest enzyme dose (3.57 U/mg), whilst run 13 demonstrated significantly lower (3.58 µmol/mL) glucose yields; however, this was after the shortest incubation time (40 min) and lowest enzyme dose (1.18 U/mg) at optimal substrate loads (0.05 g) and temperature (50 °C), suggesting that a 2-fold longer incubation time and higher enzyme dose are required for a 23.5% increase in glucose concentration (Section 3.11). However, longer incubation times and higher enzyme doses would incur higher costs; therefore, it may be beneficial for industry to implement shorter incubation times and lower substrate loads and obtain lower yields, rather than longer incubation times and higher enzyme doses that would have higher cost implications. The glucose obtained from cellulose hydrolysis may be used as a feedstock in several applications, including biofuel production. Darwesh et al. [1] utilised cellulases produced by *A. niger* MK543209, isolated from soil, comprising 8.34 U/mL endoglucanase, 7.22 U/mL exoglucanase, and 5.44 U/mL *β*-glucosidase to ferment wastepaper at 45 °C at (pH 5.5), and obtained 0.9 g/L of bioethanol. In a study by Liebmann et al. [38], 168 µmol/mL glucose was obtained after hydrolysis of wheat straw by yeast α-amylase and fermented to produce 0.6 g/L ethanol. Although the glucose concentrations obtained in runs 13 and 16 are lower than that required for biofuel production, the focus of this study was to obtain an isolate capable of producing a cellulase cocktail with favourable ratios of all three cellulases, specifically higher *β*-glucosidase titres for the hydrolysis of cellulosic material to glucose by on-site production and direct application to reduce costs associated with the use of commercial enzymes in the pulp and paper industry [11].

### 2.8. Interaction of Variables for Cellulose Hydrolysis

The relationship between the responses and the parameters for cellulose hydrolysis generated by the quadratic model and the optimum level of each variable was studied using three-dimensional (3D) response surface plots (Figure 10) where the z-axis refers to glucose yield versus any two variables, whilst the other variables are at their optimal levels. The most significant interaction plots between (X_1_) incubation time and (X_4_) enzyme dose illustrate that longer incubation periods (120 min) and higher enzyme doses (3.57 U/mg) result in the highest glucose yields (Figure 10a). Figure 10b, which displays the combined effects of (X_1_) incubation time: (X_3_) substrate load demonstrates again that longer incubation times (100–120 min) but lower substrate loads (0.03–0.05 g) result in higher glucose yields. This indicates that the *A. japonicus* VIT-SB1 cellulases require longer incubation times to enable the establishment and maintenance of interactions between the substrate and enzyme for the maximal benefits of their activity, viz., higher glucose yields [39,40]. The plot for the combined effects of (X_3_) substrate load and (X_4_) enzyme dose demonstrate that higher glucose yields are obtained at lower substrate loads (0.03–0.05 g) and higher enzyme doses (1.59–1.89 U/mL). The 3D plots indicate enzyme doses, substrate loads, and incubation times of 1.18–3.57 U/mg, 0.03–0.05 g, and 100–120 min, respectively, result in high glucose yields. Sangib et al. [41] also obtained higher glucose yields after longer incubation times and lower substrate load/enzyme dose using a microwave reactor. Another study by Campos et al. [41] obtained 27.22 µmol/mL of glucose from 0.05 g pre-treated sugarcane bagasse using a commercial cellulase complex. Although the cellulase cocktail in this study resulted in a 44% lower glucose yield, the study above used a commercial cellulase preparation and relied on the use of a microwave reactor, which would limit the reaction volume. Both of these would incur higher costs to industry compared to the use of a novel cellulase cocktail such as the crude *A. japonicus* cellulase extract which is amenable to on-site production [42,43].

The commercial cellulase cocktail with the ratio of *β*-glucosidase: endoglucanase: exoglucanase (1:0.8:0.7) was tested using the conditions specified by the manufacturer and produced 12.33 µmol/mL of glucose from 0.05 g of cellulose, 5 U/mg enzyme, at 50 °C after 120 min, whilst the *A. japonicus* VIT-SB1 cellulase cocktail yielded 15.24 µmol/mL of glucose after hydrolysis using a lower enzyme dose of 3.57 U/mg, clearly demonstrating that it compares favourably to the commercial cellulase cocktail (Sigma, St. Louis, MO, USA). As mentioned above, the synergistic action of endoglucanase, exoglucanase, and *β*-glucosidase is required for the complete hydrolysis of cellulose [44]. Due to high endoglucanase and exoglucanase titres, cellulose is converted at a rapid rate to cellobiose and cello-oligosaccharides [17]. *β*-Glucosidases are responsible for converting the cellobiose and cello-oligosaccharides to glucose; however, due to low *β*-glucosidase titres complete conversion of cellobiose to glucose is not possible, making this a rate-limiting step [44]. Therefore, the higher efficiency in cellulose hydrolysis observed for the *A. japonicus* VIT-SB1 cellulase cocktail compared to the commercial *T. reesei* cocktail (Sigma, St. Louis, MO, USA) can most likely be attributed to the different cellulase ratios [2]. In the same vein, the lower glucose yields produced by the commercial cellulase cocktail could be explained by the lower *β*-glucosidase titres relative to endoglucanases and exoglucanases [11]. To confirm this, the commercial cellulase cocktail was supplemented with 0.25 U/mg of purified *β*-glucosidase and yielded 15.38 µmol/mL glucose, signifying a 19.8% increase in glucose yield, confirming that higher *β*-glucosidases are necessary to avoid substrate inhibition and ensure higher glucose yields. Although the commercial cellulase cocktail supplemented with *β*-glucosidase produced a 1% higher glucose yield compared to the crude *A. japonicus* VIT-SB1 cellulase cocktail, application of the former will result in higher costs to industry, therefore reduction or elimination of commercial cellulases and substitution with crude cellulases like that reported in this study, especially if produced on-site, will result in an overall cost reduction [3].

## 3. Materials and Methods

### 3.1. Growth of Fungal Strain

The *Aspergillus japonicus* VIT-SB1 (*A. japonicus* VIT-SB1) strain was selected from a previous screening study as a good cellulase producer producing 0.4, 35, and 61 U/mL of *β*-glucosidases, endoglucanases, and exoglucanases with potential application in cellulose hydrolysis [45]. The fungal culture was plated onto potato dextrose agar (PDA) and incubated at 30 °C for 5 days until fungal growth was observed. In total, 2 types of long-term stocks were prepared for all isolated fungi: 25% glycerol stocks were prepared by washing fungal spores from the 5-day PDA plates and storing them at −80 °C, and mineral oil slants of mycelial growth on PDA were stored at 4 °C [45].

### 3.2. Medium and Cultivation of Extracellular Enzymes

Crude extracellular *β*-glucosidases were produced in submerged fermentation using a minimal medium comprising (g/L) (5 g) KH_2_PO_4_, (0.5 g) MgSO_4_·7H_2_O, (0.05 g) CaCl_2_. 2H_2_O, and 2% wheat bran [45]. Each 250 millilitre (mL) flask contained 50 mL of medium and 2 5 mm (millimetre) mycelial plugs of actively growing hyphae and was incubated at 30 °C for 7 days at 125 rpm (Eppendorf, New Brunswick Scientific, Incubator Shaker series, Innova 44, Gemiston, South Africa). After incubation, the cell-free supernatant was recovered by centrifuging the medium at 16,837× *g* for 10 min (Eppendorf centrifuge 5418, Gemiston, South Africa) and *β*-glucosidase activity was determined using the method described below.

### 3.3. Enzyme Assays

*β*-Glucosidase activity was quantified using the method of Kao et al. [27]. The reaction mixture included 10 microlitres (µL) of enzyme added to 10 µL of 4 millimolar (mM) *4*-nitrophenyl-*β*-D-glucopyranoside (*4*-NPG) in 50 mM sodium acetate buffer pH 5.0 and incubated at 55 °C for 5 min in a water bath and terminated by the addition of 160 µL 1000 mM Na_2_CO_3_. The absorbance was read at 410 nanometres (nm) using a spectrophotometer (Shimadzu UV1800, Kyoto, Japan). One unit of activity was defined as the amount of enzyme needed to release 1 µmol of phenol equivalents per minute at 55 °C. All experiments were duplicated, and a standard curve using *4*-nitrophenol (*4-*NP) in 50 mM sodium acetate buffer (pH 5.0) was established [28]. The Beer–Lambert equation was used to calculate enzyme activity:Enzyme activity (U/mL) = ∆AV/*ε*tv 
where A = change in absorbance, V = total volume of reaction (mL) divided by *ε* = molar extinction co-efficient of 4-NP (µM), t = reaction time (minutes), and V = volume of the enzyme (mL).

Endoglucanase and exoglucanase activities were quantified using the dinitrosalycyclic acid (DNS) method [45] with 1% avicel or carboxymethyl cellulose (CMC) as the substrate in 50 mM citrate buffer (pH 5.0), respectively. The reaction mixture contained 66.6 µL of the enzyme, and 600 µL of 1% avicel or CMC, incubated in a water bath (Labcon 5032 U, Gauteng, South Africa) for 15 min at 55 °C. The reaction was terminated by adding 1 mL of DNS to the reaction mixture; it was boiled at 100 °C for 5 min and the absorbance was read at 540 nm using a spectrophotometer (Shimadzu UV1800, Japan). One unit of activity was defined as the amount of enzyme needed to release 1 µmol of glucose equivalent per minute at 55 °C [45]. All assays were carried out in duplicate. A standard curve was established at 595 nm using a spectrophotometer (Shimadzu UV1800, Japan) and used to calculate enzyme activity.

### 3.4. Optimisation of Enzyme Production: One Variable at a Time (OVAT)

To optimise the growth parameters for optimal *β*-glucosidase production, OVAT was used to evaluate the effect of a single parameter at a time, and thereafter, manifest it as the standard condition before optimisation of the next parameter. Enzyme activity was assayed after each step to determine the optimal yield. The experiments were conducted in two independent experiments with duplicate assays from each experiment [28]. Production of endoglucanase and exoglucanase was monitored to ensure that the selected parameters were conducive to their production. The variables tested included incubation time, pH, incubation temperature, agitation, carbon, and nitrogen sources, as per the method of Singh et al. [35].

To determine the optimum incubation time for enzyme production, shake flasks consisting of 25 mL of minimal medium as per Section 3.2 were inoculated and incubated. Samples were obtained every 48 h for 14 days and enzyme activity was assayed. Optimum pH for enzyme production was determined by inoculation of minimal salt medium with different pH buffers: 50 mM sodium acetate buffer (pH 3.0 to 5.0), 50 mM sodium phosphate buffer (pH 6.0 to 8.0), and 50 mM glycine–NaOH buffer (pH 9.0–10.0) were incubated for their optimal time for production and thereafter assayed for enzyme activity. The optimum incubation temperature was determined by setting up shake flask cultures with media at the optimal pH and incubating them at different temperatures (20, 30, 40, 50, 60, 70, and 80 °C) at 125 rpm for the optimal time duration. Thereafter, the extracts were assayed for enzyme activity. To determine the optimum agitation for fermentation, broth media with optimal pH was inoculated and incubated at the optimal temperature at different agitation rates (100, 125, 150, 175, and 200 rpm), and thereafter assayed for enzyme activity.

To maximise enzyme production, various carbon and nitrogen sources were supplemented into the standard minimal medium at optimal pH discussed in Section 2.3. The following carbon (1%) lactose, fructose, glucose, cellobiose, avicel, maltose, glycerol, glucose, sucrose and wheat bran and nitrogen sources, (1%) glycine, ammonium nitrate, peptone, casein, tryptone, yeast extract, ammonium sulphate, soy peptone, and urea [4] inoculated with standardised culture and incubated under optimal conditions as described above. The concentration of the optimal carbon and nitrogen sources were determined by supplementing the minimal medium with different concentrations (0.25, 0.5, 0.75, 1.0, 1.25, 1.5, 1.75, 2%) of the optimal carbon and nitrogen sources and thereafter assaying for enzyme activity.

### 3.5. Statistical Optimisation, Experimental Design, and Data Analysis

#### 3.5.1. Plackett Burman Design (PBD)

In this study, six variables were selected for the PBD: incubation time (X_1_), pH (X_2_), incubation temperature (X_3_), agitation (X_4_), soy peptone (X_5_), and wheat bran (X_6_) (Table 4). Twelve experimental runs were tested for the six variables mentioned above represented by a high (+) and low (−) level. To confirm that a significant effect was studied, the high and low levels were equidistant from the optimal level and adequately far apart from one another.

All the experimental runs were conducted in duplicate, and the average of the results of the duplicated runs and assays are reported in Table 5. The PBD was based on the first-order polynomial model Equation (3):Y = *β*_0_ Σ*β*_i_X_i._
(3)
where Y = response (peak area and retention factor), *β*_0_= model intercept, *β*_i_ = linear coefficient, and X_i_ = level of the independent variable. To estimate the significant factors of the PBD, the data were analysed using the R Studio software Version 4.0.3 (R Core Team) to estimate the significant factors. The *p*-values and R coefficients were determined using analysis of variance (ANOVA) to evaluate the significance and fit of the regression model. After the analysis, those variables with the highest significance on *β*-glucosidase production were selected for the second level of optimisation using the Box Behnken Design (BBD) of Response Surface Methodology (RSM).

#### 3.5.2. RSM usingBox Behnken Design

The three most significant variables from PBD, pH (X_2_), incubation temperature (X_3_), and agitation (X_4_), were subjected to a three-level three-factor BBD with replicated centre points to determine their main interactions and the quadratic effects of these variables (Table 6). R studio was used for statistical analysis and design of experiments. The design comprised 3 replicates of the centre point and 16 combinations (Table 7). The average *β*-glucosidase activity was taken as the response (Y). To obtain an empirical model that relates the response to the independent variables, a multiple regression analysis of the data was carried out. The second-order polynomial Equation (4),
Y= *β*_0_ + Σ*β*_i_X_i_ + Σ*β_ii_*X_i_^2^ + Σ*β*_ij_X_i_X_j_, (4)

Where Y is the response (peak area), *β*_i_, *β*_ii,_ and *β*_ij_ are the coefficients of the linear, quadratic, and interaction terms, respectively. X*i* and X*j* are the independent variables. The average of the duplicated experiments and assays was the response for each run. The data were analysed using two-way ANOVA with Tukey’s multiple comparison tests (*p* < 0.05) using the R studio (R Core Team, R), and 3D response surface and contour plots were graphed using ggplot2 [46].

### 3.6. Scaled-Up Fermentation in Optimised Conditions

The optimised parameters from the statistically designed experiments were implemented for the scaled-up production of *β*-glucosidase. The minimal medium was prepared as described in Section 3.2 and supplemented with the optimised soy peptone and wheat bran concentrations. In total, 500 mL of medium was prepared in 2 L Erlenmeyer flasks, inoculated with 20 5 mm fungal plugs (5-day old fungal mycelium), and incubated at the optimised parameters in a shaker (Eppendorf, New Brunswick Innova 44, Gemiston, South Africa). After incubation, the mycelia-free supernatant was recovered by centrifuging the cultured medium at 16,873× *g* for 10 min (Eppendorf centrifuge 5418, Gemiston, South Africa). The *β*-glucosidase activity was determined as described in Section 3.3.

### 3.7. Effect of pH and Temperature on β-Glucosidase Activity

The optimum pH for *β*-glucosidase was determined at 55 °C for 5 min in various buffers: sodium acetate (50 mM, pH 3.0–5.0), sodium phosphate (50 mM, pH 6.0–8.0), and Glycine-NaOH (50 mM, pH 9.0–10.0) containing 4 mM *4*-NPG [32]. The optimum temperature of *β*-glucosidase was determined in sodium acetate buffer (50 mM, pH 4.0) containing 4 mM 4-NPG.

### 3.8. pH Stability and Thermostability of β-Glucosidase

The pH stability was carried out by pre-incubating the crude enzyme in sodium acetate buffers (50 mM, pH 3.0 and 5.0) for 3 h at 55 °C with aliquots sampled every 30 min [35]. Residual activity was determined using standard assay conditions. The enzyme in optimum pH buffer without incubation served as the control (100% activity). The stability of the enzyme was determined in 50 mM sodium acetate buffer (pH 4.0) by pre-incubating the enzyme at optimum temperature in the absence of *4*-NPG for 3 h with aliquots sampled every 30 min. The residual activity was determined at 65 °C for 5 min as per Section 3.3. Residual activity was determined by using the enzyme in an optimum pH buffer without incubation as the control (100% activity).

### 3.9. Sodium Dodecyl Sulfate Polyacrylamide Gels

SDS-PAGE was carried out according to the procedure by Laemmli [47]. A 12% polyacrylamide was prepared. Electrophoresis was carried out at 50 V for 3 h and the gel was stained with Coomassie Brilliant Blue [48]. The approximate molecular mass of the protein was determined from the bands that developed on the gel from the spectra multicolour broad range molecular mass markers (Thermo Scientific, Waltham, MA, USA).

### 3.10. Native Polyacrylamide Gels

Native-PAGE was used to detect all three cellulases, endoglucanases, and exoglucanases. After electrophoresis, the gel was soaked in 50 mM sodium acetate buffer (pH 5.0) for 10 min at room temperature; thereafter, it was incubated in 50 mM sodium acetate buffer supplemented with 0.1% esculin and 0.03% ferric chloride for 5 min at 55 °C for detection of *β*-glucosidases. After incubation, a black band should form, corresponding to the protein band, thus confirming *β*-glucosidase activity [49]. To detect endoglucanases and exoglucanases, 1% CMC and avicel native substrate PAGE gels were prepared, respectively. Once electrophoresis was completed, the gels were incubated in 50 mM citrate buffer (pH 5.0) at 85 °C for 20 min and thereafter stained with 0.1% Congo red solution for 30 min and destained in 1000 mM NaCl until clearance bands representing endoglucanase and exoglucanase activity were obtained.

### 3.11. Optimisation of Cellulose Hydrolysis Using the Cellulase Cocktail Produced by Aspergillus japonicus VIT-SB1

RSM using the BBD was used to study the influence of four variables on cellulose hydrolysis to statistically determine the optimum combination of reaction time, temperature, substrate loading, and enzyme dosage (Table 8). The main interactions and the quadratic effects of the variables on the enzymatic hydrolysis of cellulose were also assessed. A four factor, three-level design was applied to investigate the quadratic response surfaces and construct second-order polynomial models as described in Section 3.5.2. Each variable was coded and run at three independent levels, (−), (0), and (+) levels. The (−), (0), and (+) levels in the study were selected using optimal conditions for *β*-glucosidase, as studies report a lack in the efficiency of *β*-glucosidases in current cellulase cocktails which results in lower yields [13]. Optimal (pH 5.0) and temperature (50 °C) determined in Section 3.4 for *β*-glucosidases were used to determine the high (+) and low (−) levels, respectively, whilst the other two parameter levels were selected based on the literature [28].

The significant relationships In the model were assessed using the *p*-value and the degree of fitness of the models. The variables tested include (X_1_) incubation time, (X_2_) incubation temperature, (X_3_) substrate load (g), and (X_4_) enzyme dose (U/mg). Avicel (Sigma, St. Louis, MO, USA) was used as the cellulose substrate. The DNS assay was used to determine glucose yield after incubation.

### 3.12. Commercial Cellulase Cocktail from Trichoderma reesei

A commercial cellulase cocktail produced by *T. reesei* (Sigma, St. Louis, MO, USA) was used as a positive control for the comparison of the effectiveness of the *A. japonicus* VIT-SB1 cellulolytic cocktail. The optimal conditions provided by the manufacturer for the commercial cellulase enzyme for optimal activity were 5 U/mg enzyme in (pH 5.0) buffer at 50 °C with 2 h of incubation. Therefore, 0.05 g of microcrystalline cellulose (Sigma, St. Louis, MO, USA) and 5 U/mg of enzyme in sodium phosphate buffer (pH 5.0) were incubated at 50 °C for 2 h. To test the effectiveness of *β*-glucosidase supplementation in the cellulase cocktail, 0.25 U/mg of purified *β*-glucosidase from a previous study by Singh et al. [50] was supplemented in the commercial cellulase cocktail and incubated at optimal conditions as described above.

## 4. Conclusions

This study successfully optimised the production of *β*-glucosidase in a crude cellulase cocktail by *Aspergillus japonicus* VIT-SB1 (*A. japonicus* VIT-SB1) via statistical modelling using the Plackett Burman design and Box Behnken design in submerged fermentation producing a final ratio of 1:0.1:0.2 (*β*-glucosidase: endoglucanase: exoglucanase). The most influential independent variables were identified and optimised, resulting in a 96% increase in *β*-glucosidase production. The crude cellulase cocktail was thermophilic and acidic in nature. Optimisation of hydrolysis of microcrystalline cellulose by the native cellulase cocktail using response surface methodology to assess its efficiency revealed that efficient hydrolysis is obtained at longer incubation times and lower substrate loads and enzyme doses. The crude cellulase hydrolysed cellulose and produced higher glucose concentrations than the commercial cellulase cocktail. The supplemented commercial cellulase cocktail displayed low *β*-glucosidase titres and yielded higher glucose concentrations when supplemented with *β*-glucosidase. This indicates that the crude cellulases from the native producer will be a cheaper alternative for industrial applications. This study also showed that the approach to optimise *β*-glucosidase production was successful in obtaining a more favourable ratio of all three enzymes with higher titres of *β*-glucosidases compared to endoglucanases and exoglucanases. Future studies will include scaling up the production of cellulases produced by *A. japonicus* VIT-SB1, partial purification of the enzymes of interest, and application on pulp and paper mill sludge or other sources of agricultural waste materials.

## Figures and Tables

**Figure 1 ijms-24-09928-f001:**
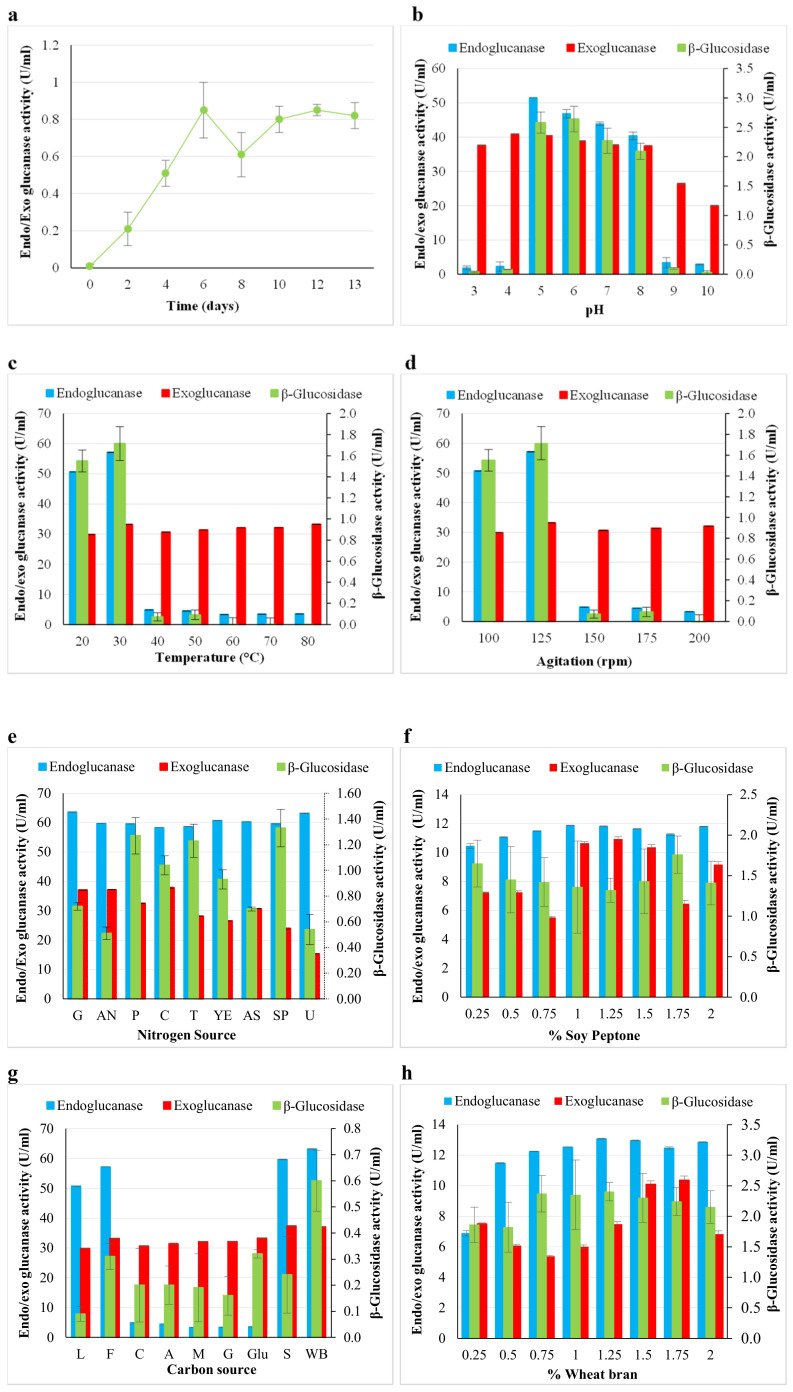
One variable at a time experiments for *β*-glucosidase production: (**a**) time course for *β*-glucosidase production (**b**), effect of pH (**c**), effect of temperature (**d**), effect of agitation (**e**), effect of nitrogen sources (G: Glycine, AN: Ammonium nitrate, P: Peptone, C: Casein, T: Tryptone, YE: Yeast extract, AS: Ammonium sulphate, SP: Soy peptone, U: Urea), (**f**) concentration of soy peptone, (**g**) effect of carbon source (L: Lactose, F: Fructose, C: Cellobiose, A: Avicel, M: Maltose, G: Glycerol, Glu: Glucose, S: Sucrose, WB: Wheat bran), and (**h**) concentration of wheat bran on the production of *β*-glucosidase, endoglucanase, and exoglucanases by *Aspergillus japonicus* VIT-SB1 in the crude enzyme extracts, (Mean ± SD, n = 4).

**Figure 2 ijms-24-09928-f002:**
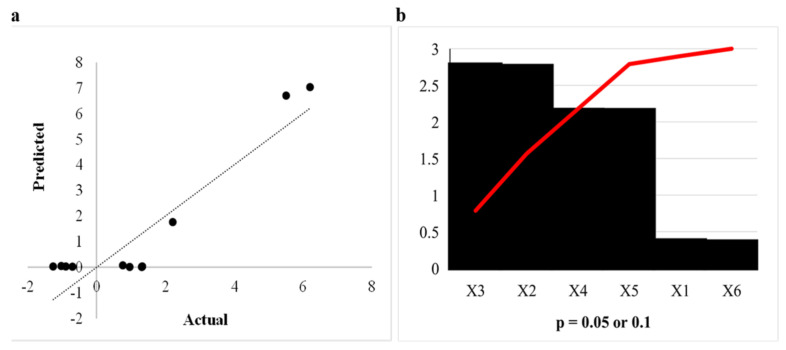
(**a**) Graphical representation of the minimal difference between the actual and predicted responses (circles) for optimal *β*-glucosidase activity, and (**b**) Pareto chart of standardised effects for the production of *β*-glucosidase for the Plackett Burman design. Incubation time (X_1_), pH (X_2_), incubation temperature (X_3_), agitation (X_4_), soy peptone (X_5_), wheat bran (X_6_).

**Figure 3 ijms-24-09928-f003:**
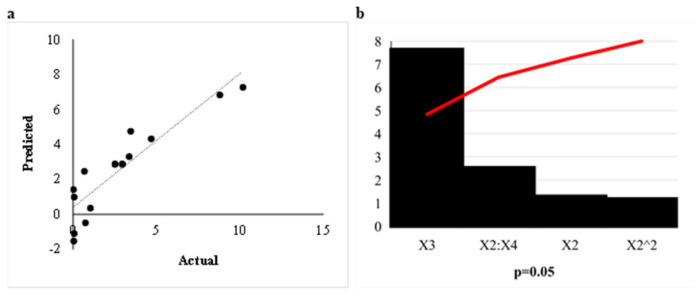
(**a**) Graphical representation of the minimal difference between the actual (straight line) and predicted responses (circles) for optimal *β*-glucosidase activity, and (**b**) Pareto chart of standardised effects for the production of *β*-glucosidase for the Box Behnken design. pH (X_2_), incubation temperature (X_3_), agitation (X_4_).

**Figure 4 ijms-24-09928-f004:**
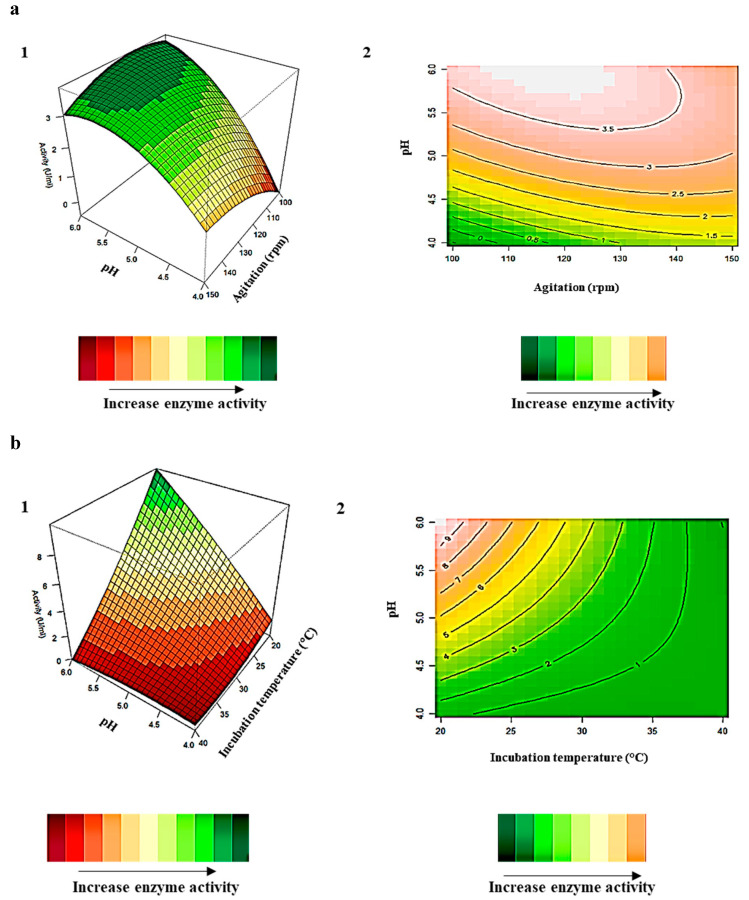
3D response surface plots (**1**) and contour plots (**2**) of the combined effects of (**a**) pH (X_2_) and agitation (X_4_), (**b**) pH (X_2_) and incubation temperature (X_3_), and (**c**) incubation temperature (X_3_) and agitation (X_4_) on *β*-glucosidase production by *Aspergillus japonicus* VIT-SB1.

**Figure 5 ijms-24-09928-f005:**
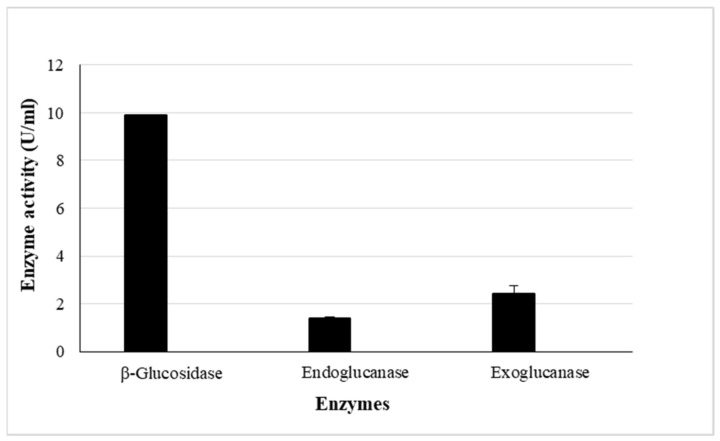
Cellulase activities in (pH 5.0) buffer at 50 °C, of the scaled-up *Aspergillus japonicus* VIT-SB1 crude extract produced under optimal conditions (Mean ± SD, n = 4).

**Figure 6 ijms-24-09928-f006:**
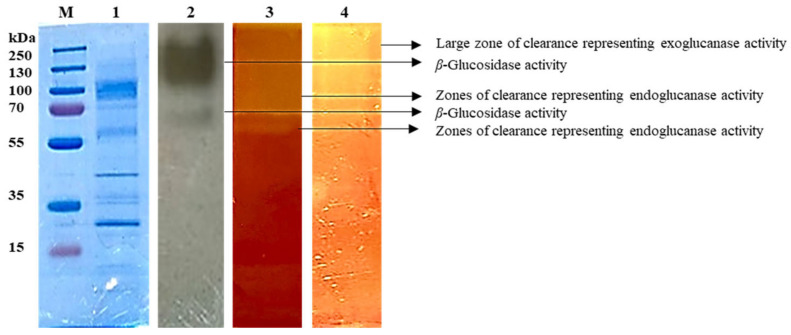
Electrophoretic analysis of *Aspergillus japonicus* VIT-SB1 *β*-glucosidases. The 12% SDS PAGE Lanes—M: Molecular mass marker (Thermo Scientific, Waltham, MA, USA), 1: crude enzyme extract, 2: crude enzyme extract displaying black precipitation, 3: crude extract on carboxymethylcellulose substrate displaying zones of clearance, 4: crude extract on avicel substrate displaying zones of clearance on Native-substrate PAGE gels.

**Figure 7 ijms-24-09928-f007:**
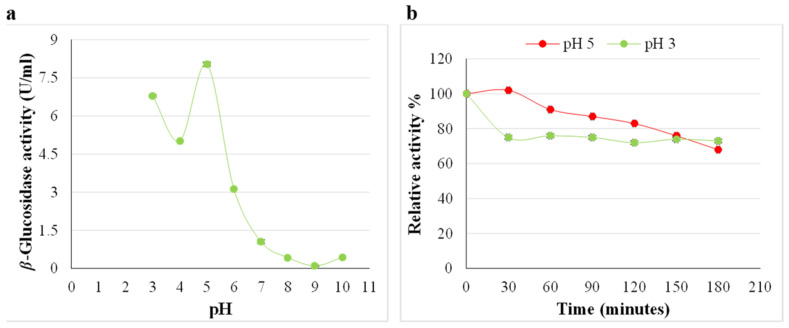
(**a**) Effect of pH on the activity of *A. japonicus* VIT-SB1 *β*-glucosidase in the crude extract (**b**) and its pH stability, (Mean ± SD, n = 4).

**Figure 8 ijms-24-09928-f008:**
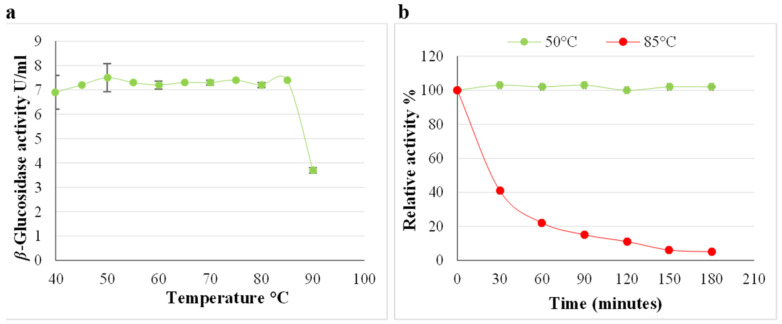
(**a**) Effect of temperature on the *Aspergillus japonicus* VIT-SB1 *β*-glucosidase activity in the crude extract (**b**) and its stability at pH 5.0, 50 °C and 85 °C, (Mean ± SD, n = 4).

**Figure 9 ijms-24-09928-f009:**
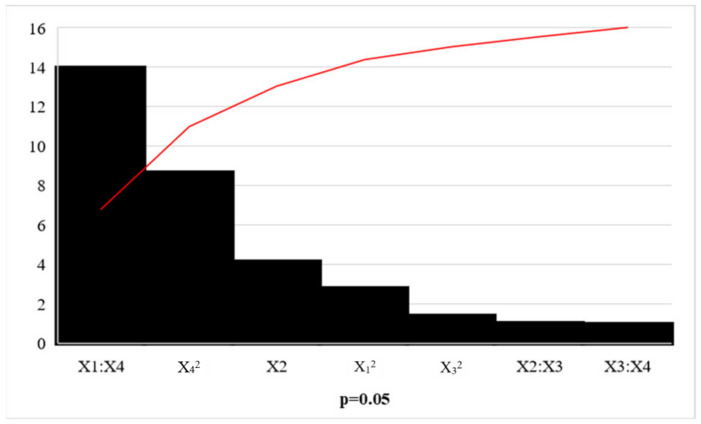
Pareto chart of standardised effects for the hydrolysis of cellulose for the Box Behnken design for incubation time (X_1_), incubation temperature (X_2_), substrate load (X_3_), and enzyme dose (X_4_).

**Figure 10 ijms-24-09928-f010:**
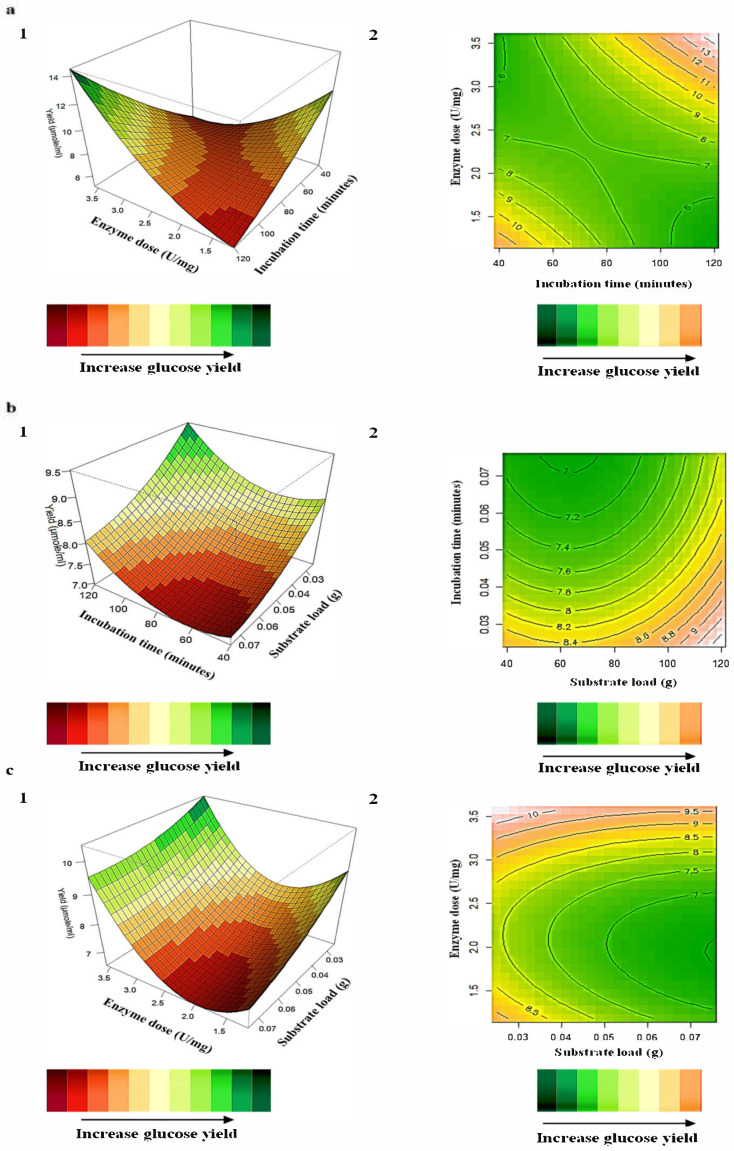
3D response surface plots (**1**) and contour plots (**2**) of the combined effects of (**a**) X_1_ (incubation time) and X_4_ (enzyme dose), (**b**) X_1_ (incubation time) and X_3_ (substrate load), and (**c**) X_3_ (substrate load) and X_4_ (enzyme dose) on glucose yield after cellulose hydrolysis.

**Table 1 ijms-24-09928-t001:** Analysis of variance (ANOVA) for six variables using Plackett Burman design for optimisation of *β*-glucosidase production by *Aspergillus japonicus* VIT-SB1.

	Df	Sum Squares	Mean Square	*F*-Value	*p*-Value
Incubation time (X_1_)	1	0.37	0.37	0.15	0.7190
pH (X_2_)	1	19.53	19.53	7.64	0.0397 *
Incubation temperature (X_3_)	1	19.79	19.79	7.74	0.0388 *
Agitation (X_4_)	1	11.98	11.98	4.68	0.0828.
Soy peptone (X_5_)	1	11.94	11.94	4.67	0.0831.
Wheat bran (X_6_)	1	0.34	0.34	0.13	0.7291
Residuals	5	12.79	2.56		

Significant *p*-values at *p* < 0.05 * and *p* < 0.1, R^2^ = 0.83. Model: *p*-value = 0.06.

**Table 2 ijms-24-09928-t002:** Analysis of variance (ANOVA) of the response surface quadratic model for the response variables for *β*-glucosidase production by *Aspergillus japonicus* VIT-SB1.

	Estimate	Standard Error	*t*-Value	*p*-Value
Model	−26.05	10.11	−2.58	0.00002 **
pH (X_2_)	3.51	2.72	1.29	0.24
Incubation temperature (X_3_)	1.75	0.23	7.64	0.0003 **
Agitation (X_4_)	−0.17	0.11	−1.59	0.16
pH (X_2_): Incubation temperature (X_3_)	−0.25	0.021	−11.67	0.00002 **
pH (X_2_): Agitation (X_4_)	0.025	0.01	2.53	0.04 *
Incubation temperature (X_3_): Agitation (X_4_)	−0.0052	0.00084	−6.17	0.0008 **
pH (X_2_)^2^	0.29	0.25	1.18	0.28
Incubation temperature (X_3_)^2^	−0.002	0.0027	−0.82	0.44
Agitation (X_4_)^2^	0.001	0.0004	2.60	0.04 *

Significant *p*-value < 0.05 *, *p* < 0.001 **, Adjusted R^2^ = 0.98. Intercept *p*-value = 0.04. Lack of fit *p*-value = 0.11.

**Table 3 ijms-24-09928-t003:** Analysis of variance (ANOVA) and regression coefficients of the response surface quadratic model for the response variables for *β*-glucosidase production by *Aspergillus japonicus* VIT-SB1.

	Estimate	Standard Error	*t*-Value	*p*-Value
Model	22.22	4.46	4.98	0.00006 **
Time (X_1_)	−0.23	0.04	−6.42	0.000002 **
Incubation temperature (X_2_)	−0.44	0.12	4.12	0.0005 **
Substrate load (X_3_)	−128.68	57.23	−2.25	0.04 *
Enzyme dose (X_4_)	−21.88	2.26	−9.67	0.000000003 **
Time (X_1_): Incubation temperature (X_2_)	−0.00002	0.0005	−0.05	0.96
Time (X_1_): Substrate load(X_3_)	−0.008	0.27	−0.03	0.97
Time (X_1_): Enzyme dose (X_4_)	0.15	0.012	13.94	0.000000000004 **
Incubation temperature (X_2_): Substrate load (X_3_)	0.73	0.73	0.99	0.33
Incubation temperature (X_2_): Enzyme dose (X_4_)	−0.002	0.03	−0.06	0.95
Substrate load (X_3_): Enzyme dose(X_4_)	16.77	17.29	0.97	0.34
Time (X_1_)^2^	0.0003	0.0001	2.79	0.01 *
Incubation temperature (X_2_)^2^	−0.004	0.0009	−5.71	0.00001 **
Substrate load (X_3_)^2^	43.11	309.44	1.39	0.18
Enzyme dose (X_4_)^2^	4.18	0.48	8.65	0.00000002 **

Significant *p*-values at Significant *p*-value < 0.05 *, *p* < 0.001 **, Model *p*-value = 0.0000000003. Adjusted R^2^ = 0.71. Lack of fit *p*-value = 0.39.

**Table 4 ijms-24-09928-t004:** Experimental variables and levels used for the optimisation of *β*-glucosidase production by the *Aspergillus japonicus* VIT-SB1in the Plackett Burman design.

			Experimental Values	
Variables	Symbol Code	Units	Low Level (−1)	High Level (+1)
Incubation time	X_1_	Days	5	7
pH	X_2_	-	4	6
Incubation temperature	X_3_	°C	20	40
Agitation	X_4_	rpm	100	150
Soy peptone (nitrogen source)	X_5_	%	1.5	2
Wheat bran (Carbon source)	X_6_	%	1	1.5

**Table 5 ijms-24-09928-t005:** Plackett Burman design matrix for the six variables used to screen for optimisation of *β*-glucosidase production by the *Aspergillus japonicus* VIT-SB1.

			Variable	Level			
Run No.	Incubation Time (Days)	pH	Incubation Temperature (°C)	Agitation (rpm)	Soy Peptone (%)	Wheat Bran (%)	Enzyme Activity (U/mL)
1	+ (7)	+(6)	−(20)	+(150)	+(2)	+(1.5)	7.03
2	+(7)	−(4)	−(20)	−(100)	+(2)	−(1)	0.02
3	−(5)	−(4)	+(40)	−(100)	+(2)	+(1.5)	0.03
4	+(7)	+(6)	−(20)	−(100)	−(1.5)	+(1.5)	1.76
5	+(7)	+(6)	+(40)	−(100)	−(1.5)	−(1)	0.02
6	+(7)	−(4)	−(20)	+(150)	−(1.5)	+(1.5)	0.03
7	−(5)	−(4)	−(20)	−(100)	−(1.5)	−(1)	0.04
8	+(7)	−(4)	+(40)	+(150)	+(2)	−(1)	0.07
9	−(5)	+(6)	−(20)	+(150)	+(2)	−(1)	6.7
10	−(5)	+(6)	+(40)	−(100)	+(2)	+(1.5)	0.01
11	−(5)	+(6)	+(20)	+(150)	−(1.5)	−(1)	0.01
12	+(7)	−(4)	+(40)	+ (150)	−(1.5)	+(1.5)	0.03

**Table 6 ijms-24-09928-t006:** Experimental codes and levels of independent variables in the Box Behnken design for optimal *β*-glucosidase production by the *Aspergillus japonicus* VIT-SB1.

		Experimental Values		
Variables	Symbol Code	Low (−)	Zero (0)	High (+1)
pH	X_2_	4	5	6
Incubation temperature (°C)	X_3_	20	30	40
Agitation speed (rpm)	X_4_	100	125	150

**Table 7 ijms-24-09928-t007:** Box Behnken design model matrix for the three significant independent variables for *β*-glucosidase production by *Aspergillus japonicus* VIT-SB1.

	Variable Level			
Run No	pH	Incubation Temperature (°C)	Agitation Speed (rpm)	Enzyme Activity (U/mL)
1	−(4)	−(20)	0(125)	0.67
2	+(6)	−(20)	0(125)	10.13
3	−(4)	+(40)	0(125)	0.04
4	+(6)	+(40)	0(125)	0.07
5	−(4)	0(30)	−(100)	0.98
6	+(6)	0(30)	−(100)	3.36
7	−(4)	0(40)	+(150)	0.77
8	+(6)	+(40)	+(150)	0.7
9	0(5)	−(20)	−(100)	3.46
10	0(5)	+(40)	−(100)	0.02
11	0(5)	−(20)	+(150)	8.81
12	0(5)	+(40)	+(150)	0.02
13	0(5)	−(30)	0(125)	2.48
14	0(5)	+(30)	0(125)	2.47
15	0(5)	+(30)	0(125)	2.95
16	0(5)	+(30)	0(125)	2.92

**Table 8 ijms-24-09928-t008:** Box Behnken design model matrix for cellulose hydrolysis using the *Aspergillus japonicus* VIT-SB1 cellulase cocktail.

	Variable Level				
Run No	Incubation Time (Minutes)	Incubation Temperature (°C)	Substrate Load (g)	Enzyme Dose (U/mg)	Glucose Yield (µmol/mL)
1	−(40)	−(35)	0(0.05)	0(2.36)	6.56
2	+(120)	−(35)	0(0.05)	0(2.36)	7.30
3	−(40)	+(65)	0(0.05)	0(2.36)	6.14
4	+(120)	+(65)	0(0.05)	0(2.36)	6.83
5	0(80)	0(50)	−(0.025)	−(1.18)	9.49
6	0(80)	0(50)	+(0.075)	−(1.18)	7.77
7	0(80)	0(50)	−(0.025)	+(3.57)	10.16
8	0(80)	0(50)	+(0.075)	+(3.57)	9.49
9	0(80)	0(50)	0(0.05)	0(2.36)	6.61
10	0(80)	0(50)	0(0.05)	0(2.36)	6.78
11	0(80)	0(50)	0(0.05)	0(2.36)	7.75
12	0(80)	0(50)	0(0.05)	0(2.36)	7.45
13	−(40)	0(50)	0(0.05)	−(1.18)	11.66
14	+(120)	0(50)	0(0.05)	−(1.18)	5.50
15	−(40)	0(50)	0(0.05)	+(3.57)	6.16
16	+(120)	0(50)	0(0.05)	+(3.57)	15.24
17	−(40)	−(35)	−(0.025)	0(2.36)	7.33
18	0(80)	+(65)	−(0.025)	0(2.36)	6.39
19	−(40)	−(35)	+(0.075)	0(2.36)	6.22
20	0(80)	+(65)	+(0.075)	0(2.36)	6.37
21	0(80)	0(50)	0(0.05)	0(2.36)	7.19
22	0(80)	0(50)	0(0.05)	0(2.36)	6.72
23	0(80)	0(50)	0(0.05)	0(2.36)	7.28
24	0(80)	0(50)	0(0.05)	0(2.36)	7.00
25	0(80)	0(50)	−(0.025)	0(2.36)	9.06
26	+(120)	0(50)	−(0.025)	0(2.36)	9.68
27	−(40)	0(50)	+(0.075)	0(2.36)	6.49
28	+(120)	0(50)	+(0.075)	0(2.36)	7.08
29	0(80)	−(35)	0(0.05)	−(1.18)	7.10
30	0(80)	+(65)	0(0.05)	−(1.18)	6.41
31	0(80)	−(35)	0(0.05)	+(3.57)	9.32
32	0(80)	+(65)	0(0.05)	+(3.57)	8.57
33	0(80)	0(50)	0(0.05)	0(2.36)	8.04
34	0(80)	0(50)	0(0.05)	0(2.36)	7.68
35	0(80)	0(50)	0(0.05)	0(2.36)	8.23
36	0(80)	0(50)	0(0.05)	0(2.36)	7.10

## Data Availability

The datasets used and/or analysed during the current study are available from the corresponding author upon reasonable request.

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
