# Peer review of "Optimisation of β-Glucosidase Production in a Crude Aspergillus japonicus VIT-SB1 Cellulase Cocktail Using One Variable at a Time and Statistical Methods and its Application in Cellulose Hydrolysis"

_ijms, 2023, doi:10.3390/ijms24129928_

Round 1

Reviewer 1 Report

The article contains interesting discoveries with direct application in practice. A large volume of experimental work has been carried out with good evidence.

I have the following comments:

The abstract is at least twice as long as allowed in the journal.

I am concerned that you call the Aspergillus japonicas -VIT-SB1 strain newly isolated, and it closely resembles the Aspergillus japonicus -VIT-SB1 strain that was isolated in 2014 (https://link.springer.com/article/10.1007/s11274 -014-1629-9).

 In your previous paper (Ref. 21), the strain was mentioned and also you gave the sequence KC128815.1 from 2015  (https://www.ncbi.nlm.nih.gov/nuccore/430768057). Please clarify the origin of the strain and describe it properly. Otherwise, it turns out that the strain was isolated twice for the first time, once in 2014 and again in 2021.

Line 18 - Obviously, Aspergillus japonicas VIT-SB1 is not newly isolated, it was isolated long ago, back in 2014. Please give the reference https://link.springer.com/article/10.1007/s11274-014-1629-9

Line 34- "mole" is wrong.

Question: how do you explain the two peaks at the pH optimum of the enzyme under study (Figure 7)?

Please check again the data in Figure 8. There is no enzyme that works regardless of temperature between 40 and 100 C, the latter temperature being boiling water, and the growth optimum for Aspergillus japonicas is 25 C. The result at 100 C seems fake.

Minor editing of English language required.

Reviewer 2 Report

1.      The name of microorganism is Aspergillus japonicus , not Aspergillus japonicas. Please correct.

2.      Authors should write the name of the microorganism culture collection from which the microorganism was bought or add a reference.

3.      In the Introduction, the authors should briefly describe the characteristics of A. japonicus, and its source, especially cellulases produced by this microorganism.

4.      The title of Figure 1 should be rewritten. First, Figure 1A presents the time course of ß-galactosidase production. Next, Figures 1B-1C. Present the effect of pH, T and agitation on endoglucanase, exonuclease and ß-glucanase activity (not only ß-glucosidase as written in the figure title). Figures 1E-H present the effect of supplements (wheat bran, peptone and soy peptone) and carbon sources on endoglucanase, exonuclease and ß-glucanase activity (not only ß-glucosidase as written in the figure title). Please also write what the abbreviations on the x-axis Figure 1G stand for.

5.      Figure 1. The same pattern bar for specific enzyme activity should be used for all figures B-H.

6.      Please improve the readability of Figure 1. It is hard to read the values in figures B-H since some bars overlap, and so much data are presented (consider using colour instead of pattern).

7.      Figure 1: On the y-axis should be written ß-galactosidase activity instead of enzyme activity (in figures B-H are three activities presented).

8.      Materials and methods: The authors should describe the experiment in which a single variable was changed at a time (Section 2.1.), including growth media composition, tested variable range, the concentration of supplements and carbon sources, conditions (e.g. T, pH, agitation, etc.)

9.      Figure 6. B and C. The is no point in shoving the molecular weight markers on native-PAGE gel. The molecular weight markers are used only on SDS-PAGE for the determination molecular weight of protein. SDS PAGE separates proteins based on their mass, while native PAGE separates proteins based on size and charge. Therefore it is impossible to determine the size of specific molecular weight marker bands and draw conclusions.

10.   Line 527: Equation for enzyme activity

Please correct the enzyme activity units; instead of „U.mL“, it should be written „U/mL“.

Reviewer 3 Report

The article proposed for review is devoted to optimizing the production of beta-glucosidase as part of a cellulase cocktail synthesized by the filamentous fungus Aspergillus japonicas VIT-SB1. The authors applied the OVAT method and two statistical methods for optimization. I would like to note that the work was done very carefully and thoughtfully. I have only a few comments on the described experiments and results.

1. When describing enzymatic activity, the authors use units/mL. In my opinion, this may distort the results because values are not normalized to the amount of protein in the sample. It is important to take into account the amount of protein in such experiments: even with small changes in cultivation conditions, a microbial cell can produce different amounts of protein, both total and specific (for example, the described enzymes).

2. In addition, the term "enzyme dose" is not clear, especially considering that it is used in units per mL. An enzyme is a protein (it can be weighed) that has a specific activity depending on the degree of purification and reaction conditions. The dose is what can be weighed. I believe that the authors should, at a minimum, recalculate the specific activity of each of the considered enzymes normalizing units per mg of total protein.

3. Figure 5: what is the difference between the enzymatic and specific activity shown in the histogram?

4. Why were specific activities for endo- and exo-glucanases also measured at 55°C and pH 5.0? Are these optimal reaction conditions for them as well? There is no information about this in the manuscript and the corresponding paper [21]. It turns out that these two enzymes are also thermostable?

5. I did not understand at what moment the salt NH4NO3 (5 g/L) in the initial minimal medium (subsections 4.2 and 4.6) was changed onto soy peptone and why?  In my opinion, 5 g per liter of nitrogen source is quite enough. Or you may vary this parameter, for instance. Why did you need to add soy peptone? Please add explanations.

6. I think that the last sentence in the Abstract about future research is not entirely appropriate.

Round 2

Reviewer 1 Report

The references in the text should not be mentioned in superscript.

The English is fine.